# Pre-Configured Error Pattern Ordered Statistics Decoding for CRC-Polar Codes

**DOI:** 10.3390/e25101405

**Published:** 2023-09-30

**Authors:** Xuanyu Li, Kai Niu, Yuxin Han, Jincheng Dai, Zhiyuan Tan, Zhiheng Guo

**Affiliations:** 1Key Laboratory of Universal Wireless Communications, Ministry of Education, Beijing University of Posts and Telecommunications, Beijing 100876, China; lixuanyu@bupt.edu.cn (X.L.); hanyx@bupt.edu.cn (Y.H.); daijincheng@bupt.edu.cn (J.D.); 2Huawei Technologies, Co., Ltd., Shenzhen 518129, China; tanzhiyuan@huawei.com (Z.T.); guozhiheng@huawei.com (Z.G.)

**Keywords:** OSD, GRAND, channel decoding, polar code

## Abstract

In this paper, we propose a pre-configured error pattern ordered statistics decoding (PEPOSD) algorithm and discuss its application to short cyclic redundancy check (CRC)-polar codes. Unlike the traditional OSD that changes the most reliable independent symbols, we regard the decoding process as testing the error patterns, like guessing random additive noise decoding (GRAND). Also, the pre-configurator referred from ordered reliability bits (ORB) GRAND can better control the range and testing order of EPs. An offline–online structure can accelerate the decoding process. Additionally, we also introduce two orders to optimize the search order for testing EPs. Compared with CRC-aided OSD and list decoding, PEPOSD can achieve a better trade-off between accuracy and complexity.

## 1. Introduction

In ultra-reliable and low latency communications (URLLC), the high reliability of short block codes becomes the key requirement [1]. To do this, cyclic redundancy check (CRC-polar codes are particularly effective [2]. For decoding short CRC-polar codes, the state-of-the-art method is CRC-aided (CA)-successive cancellation list (SCL) decoding [3].

Two cutting-edge short code decoding algorithms are ordered statistics decoding (OSD) [4] and guessing random additive noise decoding (GRAND) [5]. OSD is a decoder near the maximum likelihood (ML) and ideal for parallel design. However, the decoding complexity of *s*-order OSD can be too high to address.

Therefore, many pieces of early research have been done to reduce the complexity of OSD [6,7,8,9]. Recently, a threshold-based OSD decoder was able to reduce the number of tested codewords [10]. CA-OSD [10] and segmentation-discarding decoding [11] limit the number of valid codewords to improve performance. Probability-based OSD [12] calculates the promising probability and success probability to discard the candidate codewords.

Moreover, on the other hand, GRAND, first proposed in [5], provides a new perspective for ML decoding by estimating the noise sequence. Ordered reliability bits GRAND (ORBGRAND) [13] is proposed to improve the decoding throughput by generating possible error patterns (EPs). Its high-throughput and energy-efficient very large-scale integration (VLSI) circuit architecture is given in [14].

In this paper, we propose a new scheme called pre-configured error pattern (PEP) OSD that considers OSD from a new perspective. The main innovations and the advantages of this scheme are summarized as follows.

(1) Decoding process: Instead of concentrating on completing queries of the most reliable independent symbols [4] on Hamming balls as *s*-order OSD, we use plenty of pre-configured EPs like ORBGRAND onto the transformed information bits. Before decoding, massive EPs can be pre-configured, so the EPs can be continuously read and tested on the hard-decision bits to see if these EPs can fix the errors in the information bits of the permuted systematic polar codes. After a Euclidean distance competition of δ codewords that can pass the CRC check, the most possible result can be obtained. Due to the characteristics of CRC-polar codes, introducing the maximum number of valid codewords δ can stop the decoding early to achieve lower complexity.

(2) EP pre-configuring process: The EPs can be either pre-configured once for all kinds of codes (with different lengths or rates) to achieve higher decoding speed or dynamically generated before decoding to save the hardware resource. As optimizing the test order of the pre-configured EPs can further leverage the soft information, queries can be obviously saved. Two orders are introduced: index weight (IW) and Hamming weight (HW) order, as well as priority weight (PW) order. IW&HW relates to the error possibility of a specific EP and IW, similar to the logical weight in ORBGRAND [13], though only for the transformed information bits in this scheme; thus, the possible calculating complexity is reduced. Moreover, PW, in a quantitative relationship related to IW and HW, is designed to direct an efficient way to use the possible EPs.

The remainder of this work is structured as follows: preliminaries are provided in Section 2. The design of a PEPOSD decoder is given in Section 3. The generating theory and mechanism of PEP and testing order are given in Section 4. The simulations are evaluated in Section 5. Finally, conclusions are drawn in Section 6.

## 2. Preliminaries

### 2.1. CRC-Polar Codes

A CRC-polar code is characterized by its code length *n*, *k*-length information bits, and *m*-length CRC, denoted by [n,k+m]. For CRC-polar codes [15], the information bits are assigned to the channels with indices in the information set A, related to the more reliable subchannels, and |A|=k+m. The frozen bits, which have the default values, all zeros, are assigned to the complementary set $A^{c}$. The channel input depends on the encoding function:(1)$$f:c=u\cdot G_{n},$$
where u and c are the source and code block, respectively. The source block u consists of information bits $u_{A}$ and frozen bits $u_{A^{c}}$, then modulated into BPSK vector x. Suppose that x is transmitted over a noisy channel and the received vector y is represented as:(2)y=x+z,
where z is the additive Gaussian noise. Therefore, there is:(3)θ(y)=x⊕e,
where θ(y) denotes the hard decision sequence of the received vector and e denotes the EP where the “1” bits result in the flips of bits between the sequence sent and the hard decision of the received.

Note that the *i*-th element of a vector is expressed by []; for example, the *i*-th bit of the code is denoted by c[i].

### 2.2. OSD Algorithm

In OSD, two permutations, $\lambda_{1}$, $\lambda_{2}$, are performed over y and G before decoding. After these, the received signals $\tilde{y}$ and the hard decision $\tilde{\theta }\left(y\right)$ are all, respectively, reordered. For example, y is reordered by:(4)$$\tilde{y}=\lambda_{2}\left(\lambda_{1}\left(y\right)\right).$$

Meanwhile, the permutations and Gaussian elimination transform the generator matrix G into its systematic form $\tilde{G}$. Therefore, only the k+m most reliable positions of $\tilde{y}$ are considered.

Then, a number of tested codewords are compared to find the most likely estimate. In traditional OSD, codeword estimates are tested in the increasing order of the EP’s Hamming weights. For instance, in *s*-order OSD, codeword estimates with Hamming weight from 1 to *s* of the corresponding EP are compared. After performing inverse permutations, the best result of the codeword estimates is chosen as the output.

## 3. PEPOSD Decoder

In this section, we introduce the details of PEPOSD. The whole decoder that can generate and test the EPs in parallel and relative processes is shown in Figure 1. There are two key units in PEPOSD: the offline pre-configurator and the online EP estimator. The pre-configurator can generate and reorder all the EPs and only once for all codes. The related details are described in Section 6. Meanwhile, the EP estimator consists of three modules: pre-processor, EP tester, and validity checker.

We also summarize the decoding process in Algorithm 1. Here, we introduce the decoding process in detail.
**Algorithm 1:** PEPOSD for CRC-polar codes
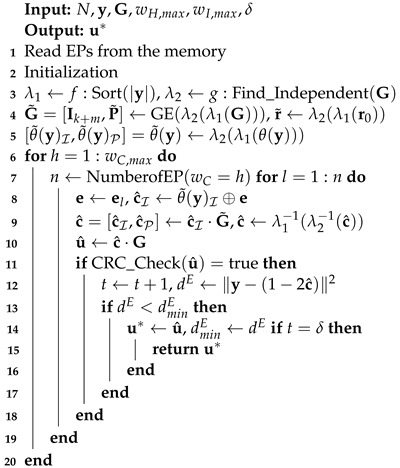


Before decoding, the signals should be preprocessed by permutations $\lambda_{1}$ and $\lambda_{2}$. Thereby, the hard decision of the signal with k+m systematic bits can be obtained. The EP tester then tests one or several EPs in parallel on the processed sequences and attains the possible result. The validity checker decides if the result can pass the CRC check. The valid results are stored in the list until the number reaches its limit δ. Otherwise, backtrack and another EP are adopted and tested. Finally, the Euclidean distances of the δ results are compared, and the most possible result is selected as the decoding output.

The pre-processor performs two permutations and the systematic transform. The first permutation $\lambda_{1}$ sorts y by its absolute value |y|, and the second permutation finds k+m linearly independent column vectors in G as the first k+m columns. Then, it performs Gaussian elimination (GE) to the permuted generator matrix $\lambda_{2}\left(\lambda_{1}\left(G\right)\right)$, so the systematic form of generator matrix $\tilde{G}$ is obtained. Thus, the generator matrix becomes $\tilde{G}=\left[I_{k+m},\tilde{P}\right]$, where $I_{k+m}$ is a (k+m)-dimensional identity matrix and $\tilde{P}$ is the parity sub-matrix.

Meanwhile, perform $\lambda_{1}$ and $\lambda_{2}$ on the hard decision θ(y) and initial index $r_{0}$, where $r_{0}$ is set by $r_{0}\left[i\right]=i$. Then, the reliability index r is obtained by $\lambda_{2}\left(\lambda_{1}\left(r_{0}\right)\right)$, which corresponds to the ascending-order index of reliability in the most reliable k+m bits. The reordered-form $\tilde{\theta }\left(y\right)$, r can be obtained. Note that $\tilde{\theta }\left(y\right)$ consists of the first (k+m) bits $\tilde{\theta }{\left(y\right)}_{I}$ and the rest $\tilde{\theta }{\left(y\right)}_{P}$, respectively, corresponding to $I_{k+m}$ and $\tilde{P}$ in $\tilde{G}$, i.e., $\tilde{\theta }\left(y\right)=\left[\tilde{\theta }{\left(y\right)}_{I},\tilde{\theta }{\left(y\right)}_{P}\right]$, where I and P denote the index set of the information and parity bits, respectively.

For each EP, the estimate of x is denoted by a codeword $\hat{c}$. The systematic bits ${\hat{c}}_{I}$ are generated by eliminating the error of hard decision $\tilde{\theta }{\left(y\right)}_{I}$,
(5)$${\hat{c}}_{I}=\tilde{\theta }{\left(y\right)}_{I}⊕e_{l},$$
where $e_{l}$ denotes the *l*-th EP. Then, the whole codeword estimate $\hat{c}$ can be calculated by:(6)$$\hat{c}={\hat{c}}_{I}\cdot \tilde{G}=\left[{\hat{c}}_{I},{\hat{c}}_{P}\right]=\left[{\hat{c}}_{I},{\hat{c}}_{I}\cdot \tilde{P}\right].$$

Therefore, a possible candidate source block $\hat{u}$ can be attained. After this, the validity checker will test if $\hat{u}$ can pass the CRC check. If the CRC check is passed, $\hat{u}$ is determined as a valid result and sent to the candidate list. Calculate the Euclidean distance $d^{E}={∥y-\left(1-2\hat{c}\right)∥}^{2}$ and compare it with the current minimum candidate $d_{min}^{E}$. If the number of candidates reaches δ, the decoding will be completed and the most likely candidate $u^{*}$ will be output. This leverages the characteristic of CRC-polar codes to control the complexity.

If the candidate is invalid or the number is not enough, come back to the EP tester and read another EP. Though the generator matrix of CRC can be calculated into the whole generator matrix, a separate check is beneficial to control the number of queries.

## 4. Pre-Configured Error Patterns

In this section, we first discuss in IW&HW order, the theoretical basis of the PEP generating mechanism. Then two integer splitting algorithms are introduced. Finally, PW order is introduced to better control the testing order of the EPs.

### 4.1. IW&HW Order

As the reliability index r is obtained by $\lambda_{2}\left(\lambda_{1}\left(r_{0}\right)\right)$, this indicates the necessary order to eliminate the errors on these bits. Upon this, referring to ORBGRAND [13], we can define reliability weight (RW), IW, and HW. The reliability weight is the sum of the approximate reliability of e, which can be calculated by:(7)$$w_{R}\left(e\right)=\sum_{i=1}^{k+m}\tilde{y}\left[i\right]\cdot e\left[i\right],$$

RW collects the reliability prior information of all permuted systematic bits. However, as RW is difficult to split and control, IW is introduced. For an error pattern e, the corresponding IW is defined as:(8)$$w_{I}\left(e\right)=\sum_{i=1}^{k+m}r\left[i\right]\cdot e\left[i\right],$$
which means the accumulation of the reliability index of all the error bits e[i] given the specific EP e. The smaller IW generally corresponds to the bigger RW, and also the more possible noise effect of the specific EP. IW gives a quantitative integer indicator to evaluate the order to test EPs. The difference between IW and logical weight [13] is that IW only consists of the information of the systematic bits, which is determined endogenously by the OSD algorithm, and, accordingly, leads to different impacts. Furthermore, $w_{I,max}$ indicates the maximum IW in all the EPs.

Similarly, the HW of a given error pattern is defined as:(9)$$w_{H}\left(e\right)=\sum_{i=1}^{k+m}e\left[i\right].$$
where $w_{H,max}$ presents the maximum HW of all the EPs. The smaller HW often leads to some more usual errors. Without ambiguity, for all eligible e, $w_{I}\left(e\right),w_{H}\left(e\right)$ are abbreviated as $w_{I},w_{H}$.

To pre-configure the EPs with all IW and HW we set, the process of PEP generation is designed as follows. We first generate EPs whose $w_{H}=$ 1. While generating “new” EPs whose $w_{H}$ is from 2 to $w_{H,max}$, the generator first reads the “old” EPs whose $w_{H}^{*}=w_{H}-1$, storing into $E_{old}$. By splitting only the biggest integer in old EPs and putting the small integers aside, corresponding new EPs can be generated. The algorithm is summarized in Algorithm 2. While splitting the integer, *b* stands for the biggest number and $a_{1}$, $a_{2}$, …, and *b* are in ascending order. Thus, all EPs needed can be pre-configured. An integer-splitting algorithm for ORBGRAND [13] can also be referred to.

There is an example for $w_{H,max}=4$ and $w_{I}=10$ shown in Figure 2. First, the EP with $w_{H}=1$ is generated. Then, 10 is divided into {9,1}, ⋯, {6,4}, and 4 EPs with $w_{H}=2$ are obtained. After that, 9 in {9,1} can be divided into {7,2}, {6,3}, and {5,4}, whereas 8 in {8,2} can be divided into {5,3}; thus, 4 EPs with $w_{H}=3$ are obtained. Finally, one EP with $w_{H}=4$ is generated by dividing 7 in {7,2,1} into {4,3}.

The PEP pre-configurator can produce all EPs stored in the memory before decoding numerous codes, so the decoder can continuously read EPs to significantly reduce the decoding delay, and only once is enough for all kinds of codes and all code blocks. On the other hand, while decoding a small number of codes, each EP can also be dynamically generated just before being tested to ensure better energy efficiency.
**Algorithm 2:** Generate PEPs
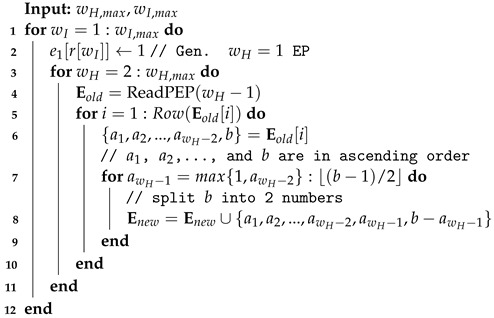


### 4.2. PW Order

As IW and HW are introduced and all the EPs have been pre-configured, PW can be defined by:(10)$$w_{P}\left(e\right)=w_{I}+\alpha \cdot {\left(w_{H}\right)}^{\beta }.$$
where α and β are parameters to be set. The order of using the EPs depends on their PW, which indicates a special order to prevent the decoder from trying some EPs with a super low possibility even if its HW is small.

Figure 3 gives a hypothetical example to see the difference of the PEPOSD scheme between IW&HW order and PW order. Figure 3a shows the IW&HW-order PEPOSD. The decoder first tests the $w_{H}=1$ EPs in the order of $w_{I}$. After that, it tests those with $w_{H}=2$, then 3, and so on. Meanwhile, in our new proposed scheme, the decoder just tests the EPs in the order of PW. Figure 3b shows how the PWs of the EPs correspond to their HW and PW. Therefore, the EPs are tested from $w_{P}=2$, to $w_{P}=23$. Obviously, the two EPs with $w_{P}=7$ are tested together, and also for $w_{P}=8,9,10,15$. In this way, the order of using EPs can be optimized and some less probable EPs will be tested far later.

## 5. Performance Evaluation

In this section, for CRC-polar codes, we, respectively, compare the performance of PEPOSD with 3-order CA-OSD and CA-SCL (L=32).

### 5.1. BLER Analysis

First, we compare the BLER performance with low complexity of these algorithms. Figure 4 shows the BLER comparison between PEPOSD (IW&HW) and CA-SCL with different rates with the code length n=64 and CRC length m=6. In this figure, there is (IW/HW/δ)=(75/4/20) for PEPOSD. This demonstrates that PEPOSD outperforms CA-SCL by about 0.3 dB with the close complexity for the high rates. Increasing the CRC length can improve the performance of PEPOSD while this worsens CA-SCL, so the advantage can be more obvious.

Figure 5 shows the BLER comparison with different code rates from 0.5 to 0.85 among PEPOSD, CA-OSD, and CA-SCL when $E_{b}/N_{0}=4.0$. This shows that PEPOSD${}_{2},\left(75/4/20\right)$ can achieve close accuracy with CA-OSD. Moreover, when R=0.5 and R=0.68 or higher, PEPOSD outperforms CA-SCL obviously. More detailed analysis about PEPOSD related to its complexity is given in Section 5.2.

Then, we analyze the ultimate performance with higher complexity and bigger code length. Figure 6 shows the performance comparison for the [128,108+11] CRC-polar code. PEPOSD(IW&HW) is here with (IW/HW)=(100/4) and different δ. Meanwhile, PEPOSD(PW) with (IW/HW/δ/α/β)=(100/4/1/3/2) and CA-SCL (L=32) are shown. The average decoding time is obtained from the same CPU. It can be concluded that PEPOSD can achieve better performance at a high rate for 128-bit CRC-polar codes and the decoding complexity can also be smaller than CA-SCL in high SNR areas. Also, another decoding algorithm for polar codes with lower complexity, CA priority-first SC, is introduced and simulated. Though it can decode faster in higher SNR areas, PEPOSD shows much better reliability and can decode faster in the lower SNR areas. Moreover, the PW order performs better in both BLER performance and decoding time for this code. The results show the advantage of the proposed scheme.

To enrich the results of the performance of PEPOSD, the BLER comparisons for n=102 and n=112 are given in Figure 7. It is obvious that PEPOSD, especially in the PW order, can obtain a good trade-off between accuracy and complexity.

In addition, the settings of α and β in the PW-order PEPOSD depend on some simulation results. Figure 8 shows for the [128,108+11] code, when $E_{b}/N_{0}=4.0\;\mathrm{dB},\left(\mathrm{IW}/\mathrm{HW}/\delta \right)=\left(100/4/1\right)$, the effects on BLER and average flips of different settings of α and β. It is obvious that α=3,β=2 is the best setting in such a circumstance, and this is why we choose this.

Therefore, the simulation results show that PEPOSD achieves a better trade-off between accuracy and complexity than CA-OSD, and also can perform better for some short codes than CA-SCL. Moreover, the parameters can be configured flexibly and the decoding process can be parallelized to further increase its throughput.

### 5.2. Complexity Analysis


First, we compare the computational complexity of the proposed scheme with CA-SCL. Specifically, referring to other papers about OSD, as the other complexity introduced by some pre-processing operations is too small to calculate, it should be noted that all operations calculated below are modulo-two operations (XOR) in this scheme, so the hardware resources and time spent will be obviously less with the same quantity in the engineering practice and hardware implementations. As most of the OSD research does, we focus on the queries needed in the decoding process, and also the number of the EPs tested. Therefore, the number of bit flipping in this period can be calculated by $\sum_{i=1}^{Q}w_{H}\left(e\left[i\right]\right)$, where *Q* denotes the queries. Another key complexity that we consider compared with SCL, GRAND, or other algorithms is GE in the pre-processor, of which the complexity can be calculated by $O(n\cdot {\left(min(k,n-k)\right)}^{2})$. Moreover, there are some parallel or other efficient implementations that can optimize the process, like in [16].

Also, multiplication and addition operations needed in CA-SCL can be expressed as O(n·L·log(n)). Thus, Table 1 displays the complexity estimation of PEPOSD, CA-OSD, and CA-SCL. The queries of PEPOSD are mainly based on δ, if IW and HW are relatively high enough. In conclusion, PEPOSD can obviously achieve lower complexity for high-rate codes, and for lower rates, PEPOSD may outperform CA-SCL as it is with modulo-two operations, which needs more hardware analysis to prove.

Observing together with Figure 5, it is obvious that PEPOSD${}_{2},\left(75/4/20\right)$ can achieve close accuracy with CA-OSD, and its average number of bit flipping is 1/9 to 1/36 of 3-order CA-OSD. Also, PEPOSD${}_{3},\left(100/4/100\right)$ can obtain better accuracy than CA-OSD and the queries can be greatly reduced at the same time. For high-rate codes, PEPOSD can outperform CA-SCL and CA-OSD both in accuracy and complexity.

Finally, as PW is introduced, the queries reduction of the schemes with the IW&HW and PW orders is compared in Table 2. For the [64,46+6] and [128,108+11] CRC-polar codes, using PEPOSD with (IW/HW/δ)=(100/4/1), the number of queries is reduced by 10–30%.

## 6. Conclusions

In this paper, we introduce the PEPOSD algorithm to enhance the performance of short CRC-polar codes. It integrates the generating mechanism of noise queries in ORBGRAND to the generation of error patterns in OSD. Therefore, all the EPs can be pre-generated to allow the pipeline decoding for better speed. Also, early stop by CRC check can significantly reduce the complexity.

To optimize the decoding order of the proposed scheme, two options are introduced. The IW&HW order is suitable for most circumstances, whereas PW shows lower complexity with bigger IW. In this way, the range of error patterns can be more controllable than *l*-order CA-OSD.

The simulation results show that there are several advantages in the performance and complexity of PEPOSD compared with CA-OSD and CA-SCL for CRC-polar codes, which shows a promising prospect.

## Figures and Tables

**Figure 1 entropy-25-01405-f001:**
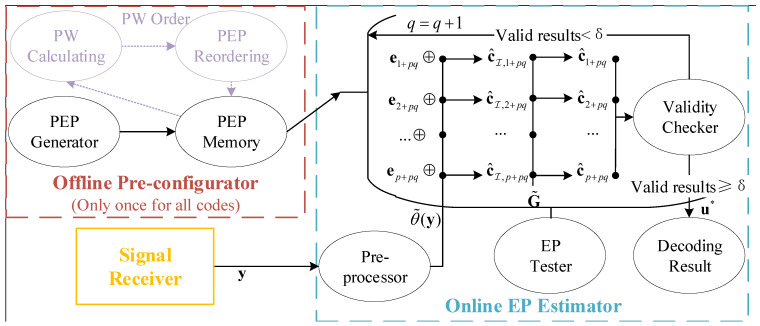
The offline–online structure of a PEPOSD decoder.

**Figure 2 entropy-25-01405-f002:**
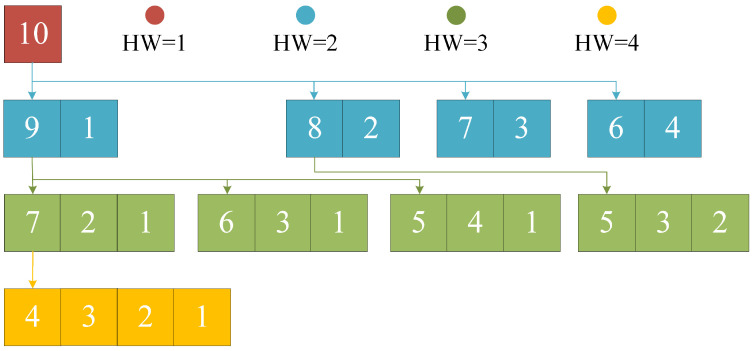
An example of generating for $w_{H,max}=4$ and $w_{I}=10$.

**Figure 3 entropy-25-01405-f003:**
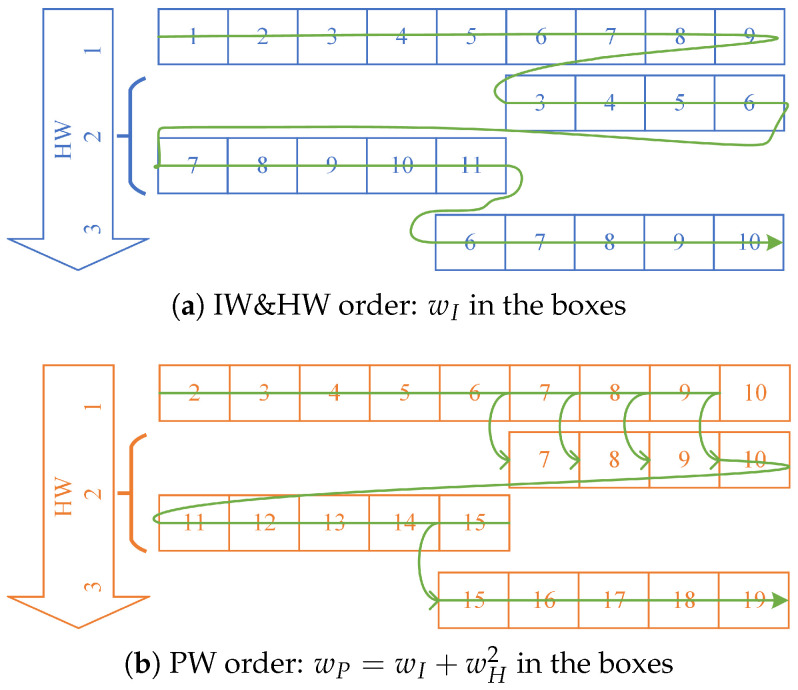
The sketches of IW&HW order and PW order.

**Figure 4 entropy-25-01405-f004:**
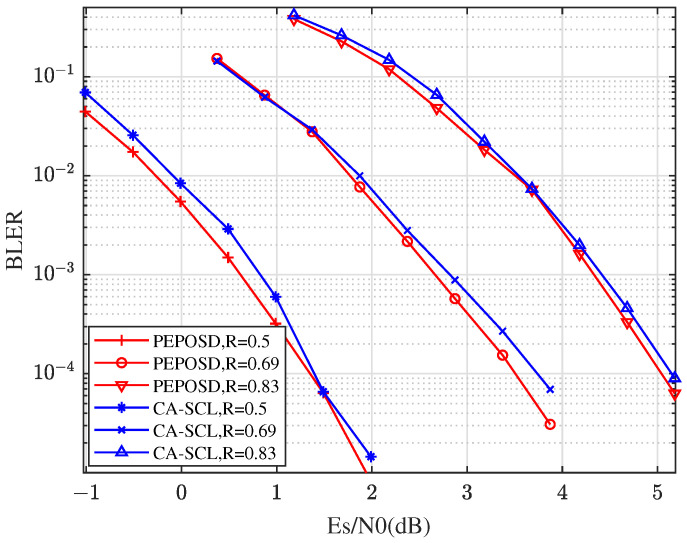
The comparison of BLER performance between PEPOSD and CA-SCL with different rates with the code length n=64.

**Figure 5 entropy-25-01405-f005:**
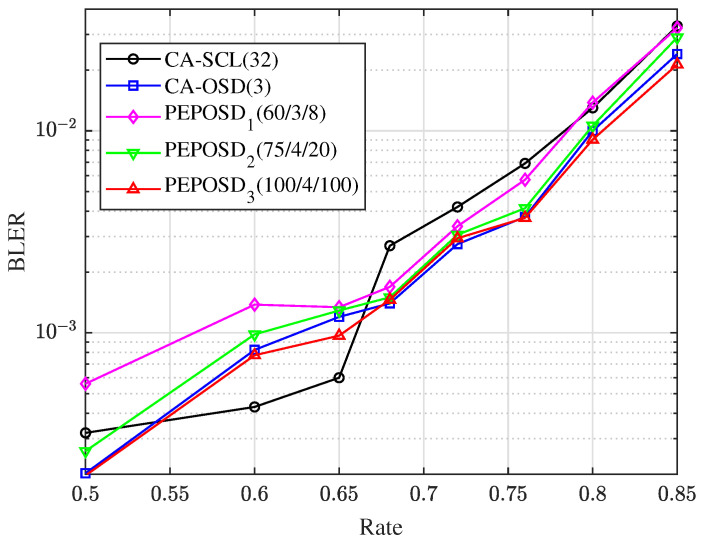
When SNR is 4.0 dB, the BLER comparison with different code rates among PEPOSD, CA-OSD, and CA-SCL.

**Figure 6 entropy-25-01405-f006:**
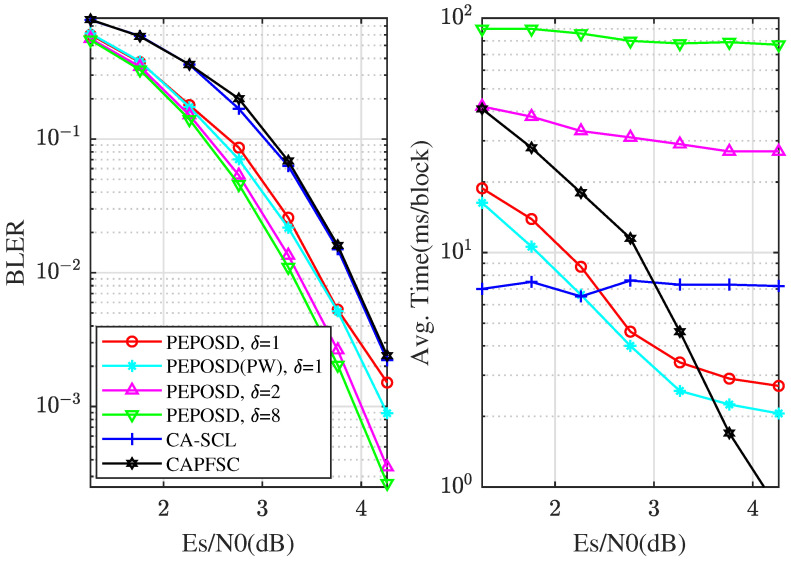
For [128,108+11] code, the comparison of BLER and average decoding time performance among different algorithms.

**Figure 7 entropy-25-01405-f007:**
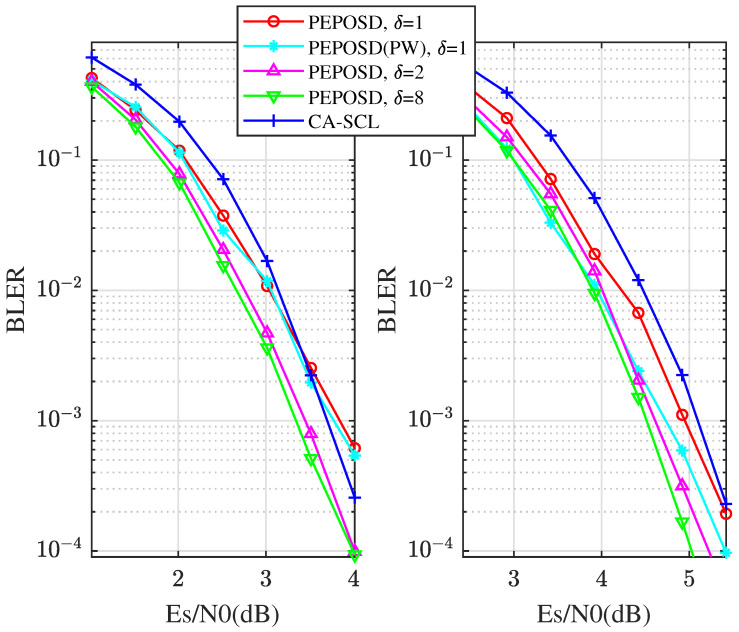
For [128,102+11] and [128,112+11] code, the BLER comparison of among different settings.

**Figure 8 entropy-25-01405-f008:**
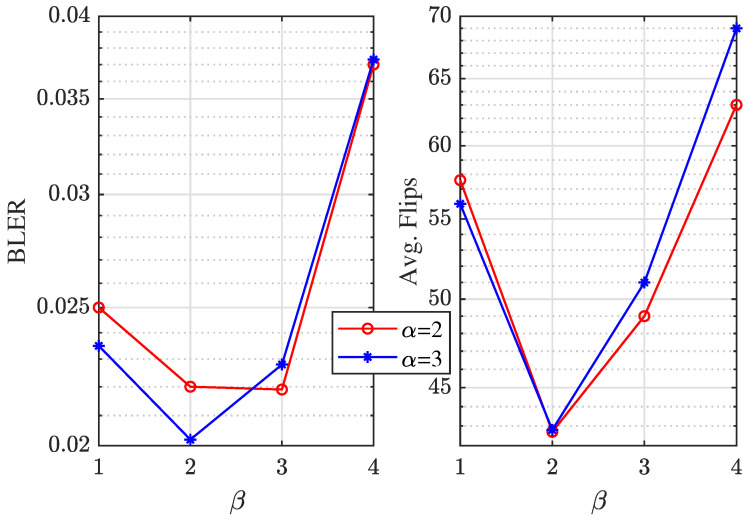
For [128,108+11] code, when $E_{b}/N_{0}=4.0\;\mathrm{dB}$, the comparison of BLER and average flips performance among different settings.

**Table 1 entropy-25-01405-t001:** Complexity estimation of PEPOSD, CA-OSD, and CA-SCL with different CRC-polar codes. For GE and CA-SCL, the number denotes the needed operations. For OSD algorithms, the number denotes the number of bit flipping.

Code	GE ^*^	PEPOSD${}_{1}$ ^**^	PEPOSD${}_{2}$	PEPOSD${}_{3}$	CA-OSD	CA-SCL
a. [64,32+6]	43,264	898	2975	18,535	8436	12,288
b. [64,44+6]	12,544	887	2722	18,159	19,600	12,288
c. [64,53+6]	1600	899	2620	18,108	32,509	12,288
d. [128,108+11]	28,672	16	103	844	273,819	10,368

* Gaussian Eliminate is necessary in all OSD algorithms, which is mod-2 operation. ** For PEPOSD${}_{1}$, PEPOSD${}_{2}$, PEPOSD${}_{3}$: n=64,δ = 8, 20, 100; *n* = 128, δ = 1, 2, 8. For PEPOSD${}_{1}$, PEPOSD${}_{2}$, PEPOSD${}_{3}$: $IW/HW=50/3,E_{s}/N_{0}=5.0$ dB, n=64,δ=8,20,100;n=128,δ=1,2,8.

**Table 2 entropy-25-01405-t002:** The average queries of IW&HW and PW order with (IW/HW/δ)=(100/4/1) for the CRC-polar codes.

Order	SNR = 2.0 dB ^1^	SNR = 2.5 dB	SNR = 3.0 dB	SNR = 3.5 dB	SNR = 4.0 dB
(a) n=64	k=46	m=6			
IW&HW	23.1	11	6.8	3.9	2.1
PW	16.5	11	5.9	3.5	2.1
(b) n=128	k=108	m=11			
IW&HW	925	641	275	101	25
PW	928	514	188	75	22

^1^ SNR denotes $E_{b}/N_{0}$.

## Data Availability

No new data were created.

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
