# Peer review of "Pre-Configured Error Pattern Ordered Statistics Decoding for CRC-Polar Codes"

_entropy, 2023, doi:10.3390/e25101405_

Round 1
Reviewer 1 Report
In the paper, the authors propose a modification of the CRC-Aided ordered statistics decoding (CA-OSD) for short CRC-polar codes.
Inspired by ordered reliability bits guessing random additive noise decoding (ORBGRAND), the authors propose to use pre-configured error patterns (EPs) to see if these can fix the errors information bits of the permuted systematic polar code. This leads to lower decoding complexity.
To save queries in decoding, two orders are proposed: index weight (IW) and Hamming weight (HW) order and priority weight (PW) order. PW involves two parameters denoted $\alpha$ and $\beta$.
Simulations show that the proposed PEPOSD algorithm has slightly better error rate performance compared to CA-successive cancellation list decoding (CA-SCL) and lower complexity compared to CA-SCL and CA-OSD algorithms.
The paper is clearly and well written. It is interesting for the polar codes research community.
Some issues detected by me in the paper are:
1) In Algorithm 1 (page 4), variables $t$ and $d_{\min}^E$ have to be initialized before the "for" cycle beginning at line 5. Also, at line 7: instead of "$\ominus \textbf{e}$" --> "$\oplus \textbf{e}$".
2) In Section 5, values $\alpha = 2$ and $\beta = 3$ are used for the PW in PEPOSD(PW) algorithm, but no discussion is given for how choosing these values. Please explain the choosing of $\alpha$ and $\beta$ in PEPOSD(PW) algorithm.
3) On page 7, in the label of Figure 5: "different code rate" --> "different code rates".
Author Response
Thanks for your kind suggestions!
The mistakes that you mentioned have been modified. Also, the choice of alpha and beta are decided by simulation results, so we add the simulation results and the reasons in red. In addition, Fig 8 illustrates the advantages of the chosen parameters.
Reviewer 2 Report
The paper introduces a GRAND-like decoding method for polar codes with CRC. The proposed approach appears to be rather straightforward, and it seems to be applicable to any linear block code, not just polar codes.
Detailed comments:
1. It should be noted that GRAND was initially proposed in the introduction of [6].
2. Please provide performance and complexity comparison of the proposed approach with at least one of the algorithms introduced in [7,8,9]
3. The authors claim: "Specifically, it should be noted that all operations are modulo-two operations (XOR) in this scheme". This is definitely not true, since you need to compare LLRs to identify the most reliable basis, and you need to compare the scores of different codewords to select the best one. Please make sure that these operations are properly accounted.
4. Please provide performance and complexity comparison with low-complexity decoding algorithms specifically designed for polar codes, e.g. sequential decoding
NA
Author Response
Thanks for your kind advice!
The first and the third comments are absorbed, and we give some explanations on them.
For the second suggestion, the older variants of OSD are not suitable to be directly applied to CRC-polar codes. Study on it can be interesting, but it's a little bit over the range of the paper of decoding polar codes, and as the revision time is limited, we are sorry that it's difficult to provide this results.
For the last suggestion, we added the comparison with CA priority-first SC in Figure 6. Though it can achieve super low complexity in the high SNR areas, PEPOSD shows much better BLER performance. Thanks for your comment!